



# The Regional Climate-Chemistry-Ecology Coupling Model RegCM-Chem (v4.6)-YIBs (v1.0): Development and Application

Nanhong Xie[1], Tijian Wang[1]*, Xiaodong Xie[2], Xu Yue[2], Filippo Giorgi[3], Qian Zhang[1], Danyang Ma[1], Rong Song[1], Baiyao Xu[1], Shu Li[1], Bingliang Zhuang[1], Mengmeng Li[1], Min Xie[1], Natalya Andreeva Kilifarska[4], Georgi Gadzhev[5], Reneta Dimitrova[6]

[1]School of Atmospheric Sciences, Nanjing University, Nanjing, 210023, China

[2]School of Environmental Sciences and Engineering, Nanjing University of Information Science and Technology, Nanjing, 210023, China

[3]Earth System Physics Section, the Abdus Salam International Centre for Theoretical Physic, Trieste, 34100, Italy

[4]Climate, Atmosphere and Waters Research Institute, Bulgarian Academy of Sciences, Sofia, 1113, Bulgaria

[5]National Institute of Geophysics, Geodesy and Geography, Bulgarian Academy of Sciences, Sofia, 1113, Bulgaria

[6]Department of Meteorology and Geophysics, Faculty of Physics, Sofia University, Sofia, 1113, Bulgaria

*Corresponding to*: Tijian Wang (tjwang@nju.edu.cn)

**Abstract.** The interactions between the terrestrial biosphere, atmospheric chemistry, and climate involve complex feedbacks that have traditionally been modeled separately. We present a new framework that couples the Yale Interactive terrestrial Biosphere (YIBs), a dynamic plant-chemistry model, with the RegCM-Chem model. RegCM-Chem-YIBs integrates meteorological variables and atmospheric chemical composition from RegCM-Chem with land surface parameters from YIBs. The terrestrial carbon flux calculated by YIBs, are fed back into RegCM-Chem interactively, thereby representing the interactions between fine particulate matter ($PM_{2.5}$), ozone ($O_3$), and carbon dioxide ($CO_2$). For testing purposes, we carry out a one-year simulation (2016) at a 30 km horizontal resolution over East Asia with RegCM-Chem-YIBs. The model accurately captures the spatio-temporal distribution of climate, chemical composition, and ecological parameters. In particular, the estimated $O_3$ and $PM_{2.5}$ are consistent with ground observations, with correlation coefficients (R) of 0.74 and 0.65, respectively. The simulated $CO_2$ concentration is consistent with observations from six sites (R ranged from 0.89 to 0.97) and exhibits a similar spatial pattern when compared to carbon assimilation products. RegCM-Chem-YIBs produces reasonably good gross primary productivity (GPP) and net primary productivity (NPP), showing seasonal and spatial distributions consistent with satellite observations, and mean biases (MBs) of 0.13 and 0.05 kg C m$^{-2}$ year$^{-1}$. This study illustrates that the RegCM-Chem-YIBs is a valuable tool to investigate coupled interactions between the terrestrial carbon cycle, atmospheric chemistry, and climate change at a higher resolution in regional scale.



## 1 Introduction

Air pollution and climate change are major focal points in atmospheric and environmental science (Hong et al., 2019; Kan et al., 2012). In this respect, China exhibits both high air pollution levels and large greenhouse gas emissions (Zheng et al., 2018; Li et al., 2016a). The consequences of China's air pollution on global, regional, and urban climate are significant (Liu et al., 2022; Lu et al., 2020). Conversely, global warming impacts the dynamics, physics, and chemical mechanisms underlying atmospheric pollutant formation, underscoring a robust link between atmospheric chemistry and climate change (Baklanov et al., 2016; Fiore et al., 2015; Fiore et al., 2012).

$PM_{2.5}$, $O_3$, and $CO_2$ are important for regional air pollution and climate. $O_3$, a potent pollutant, is harmful for human health and can also harm chloroplasts in plant cells, consequently influencing the carbon assimilation efficiency of land ecosystems (Xie et al., 2019; Ainsworth et al., 2012). Similarly, $PM_{2.5}$ is not only one of the most dangerous pollutants for human health (Kim et al., 2015), but also affects atmospheric radiation mechanics, modulates radiation fluxes reaching vegetation canopies, and hence impacts plant physiological processes and terrestrial carbon fluxes (Lu et al., 2017; Strada and Unger, 2016). Terrestrial ecosystems, absorbing nearly 30% of anthropogenic $CO_2$ emissions, play an essential role in the global carbon cycle, for which even minor alterations can trigger significant oscillations in atmospheric $CO_2$ concentrations, potentially destabilizing the global climate (Forkel et al., 2016; Ahlstrom et al., 2015). As a result, $PM_{2.5}$, $O_3$, and $CO_2$ exhibit intricate interplays.

Models that couple climate and chemistry are vital tools for investigating the interplay between environmental pollution and climate warming (Dunne et al., 2020; Yahya et al., 2017), and in particular the direct and indirect influences of aerosols, $O_3$, and greenhouse gases on climates at different scales (Chutia et al., 2019; Pu et al., 2017; Li et al., 2017a). For example, the Atmospheric Chemistry and Climate Model Intercomparison Project (ACCMIP) addresses this issue through the use of a range of global coupled climate-chemistry models (Young et al., 2013; Shindell et al., 2013; Lamarque et al., 2013). In fact, China has achieved significant advancements in atmospheric chemistry and coupled climate models during recent years, both at the global and regional scale. Representative models encompass BCC_AGCM2.0_CAM, BCC-AGCM_CUACE2.0, RIEMS-Chem, and RegCCMS.

BCC_AGCM2.0_CAM was coupled by the China Meteorological Administration through direct integration of the National Climate Center's atmospheric circulation model (BCC-AGCM) with the Canadian aerosol model (CAM) (Zhang et al., 2012). Atmospheric model BCC-AGCM2.0 was developed by the National Climate





Center. For example, at the regional scale the Institute of Atmospheric Physics of the Chinese Academy of Sci-
ences, has constructed the Regional Integrated Environmental Modeling System (RIEMS), which is widely used
in studies on East Asian regional climate change and severe weather systems (Scheuch et al., 2015; Xiong et al.,
2009). It incorporates atmospheric chemistry and aerosol dynamics into the Regional Integrated Environment
Modeling System and produces online simulations of meteorological parameters, aerosol chemical composition,
optical characteristics, radiation forcing, and aerosol-induced climate feedback (Li et al., 2014; Li et al., 2013a;
Han et al., 2012).

The Nanjing University developed the Regional Climate Chemistry Modeling System (RegCCMS), a syn-

thesis of the regional climate model RegCM2 and the tropospheric atmospheric chemistry model TACM, pri-
marily oriented toward investigating the spatio-temporal distribution, radiation forcing, and climatic effects of
tropospheric $O_3$ and sulfate aerosols. Subsequently, RegCM3 was coupled with TACM, integrating modules for
aerosols into RegCCMS (Zhang et al., 2014; Li et al., 2009). The system incorporates parameterization schemes
facilitating the simulation of aerosols' direct, indirect, and semidirect climatic effects. Extensive evaluations
have been carried out regarding major aerosol impacts on the meteorology and regional climate within East Asia
(Zhuang et al., 2013; Zhuang et al., 2011; Wang et al., 2010). Subsequently, Shalaby et al. (2012) developed the
regional climate-chemistry model RegCM-Chem, by coupling the CBM-Z gas phase chemistry module to ver-
sion 4 of the RegCM system, RegCM4 (Giorgi et al., 2012). RegCM-Chem also includes a simplified aerosol
scheme including radiatively interactive sulfates, carbonaceous aerosols, sea salt, and desert dust (Zakey et al.,
2006; Solmon et al., 2006), and it has been used for a variety of applications in different domains.

By developing the regional climate-chemistry-ecology model RegCM-Chem-YIBs, in which the interactive

biosphere model YIBs is coupled to RegCM-Chem. The model can produce multi-process simulations of re-
gional climate, atmospheric chemistry, and ecology, especially $PM_{2.5}$, $O_3$, and $CO_2$, and their interactions with
atmospheric variables (Xu et al., 2023; Ma et al., 2023b; Ma et al., 2023a; Xu et al., 2022; Gao et al., 2022; Xie
et al., 2020). Here we expand on these previous studies. We carry out a one-year simulation (2016) at a 30 km
horizontal resolution over East Asia with RegCM-Chem-YIBs and conduct a comprehensive assessment. We
validate the simulation not only in terms of atmospheric variables but also in terms of atmospheric composition
and ecological parameters, by comparison with a range of observations available for this period.

The paper is organized as follows. In section 2 we first describe the RegCM-Chem-YIBs system, focusing

in particular on the newly implemented coupling with the ecological component. We also describe the observa-



tion datasets used in the model assessment. The simulations are then analyzed in section 3, while section 4 pre-
sents our conclusions and a general discussion of our results and future developments.
**2 Model and Methods**
**2.1 Overall Framework**
In RegCM-Chem-YIBs, the atmospheric variables produced by RegCM (temperature, humidity, precipita-
tion, radiation, etc.) and atmospheric chemical compounds, such as $O_3$ and $PM_{2.5}$, produced by the chemis-
try/aerosol module are input into YIBs, which simulates the physiological processes of vegetation (such as pho-
tosynthesis, respiration, etc.), and calculates land process variables such as $CO_2$ fluxes, BVOC emissions, and
stomatal conductance. The output from YIBs is then fed back to RegCM-Chem, which adjusts the $CO_2$, $O_3$, and
$PM_{2.5}$ concentrations and their radiative and microphysical effects on the meteorological fields in the lower at-
mosphere, thereby achieving a full coupling between climate, chemistry, and ecology. Figure 1 shows the basic
framework of the RegCM-Chem-YIBs coupled model.

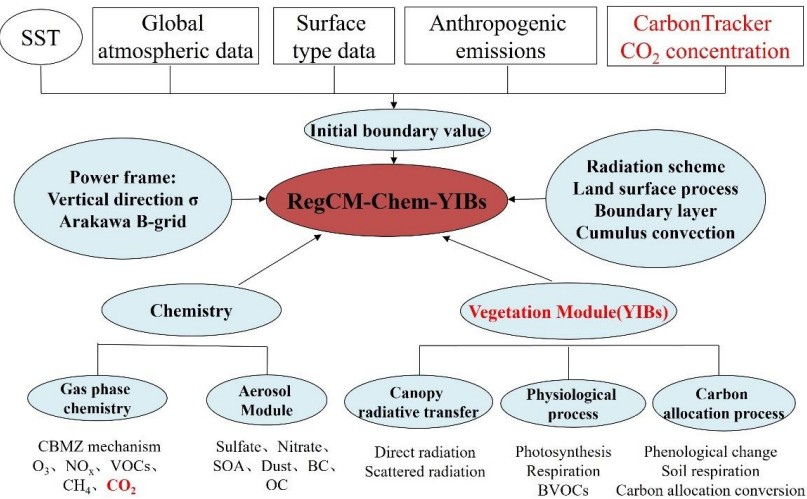


**Figure 1.** RegCM-Chem-YIBs Coupling Model Framework
**2.2 Descriptions of the RegCM-Chem model**
The inception of the RegCM system traces back to the late 1980s and early 1990s, when NCAR's (U.S. Na-
tional Center for Atmospheric Research) RegCM 1 was first developed for climate downscaling (Giorgi, 1990;



Giorgi and Bates, 1989; Dickinson et al., 1989). After a series of developments, subsequent versions were in-
troduced, such as RegCM2 (Giorgi et al., 1993), RegCM2.5 (Giorgi and Mearns, 1999), RegCM3 (Pal et al.,
2007), RegCM4 (Giorgi et al., 2012). The RegCM system presently managed, maintained, and expanded by the
Earth System Physics (ESP) section of the Abdus Salam International Center for Theoretical Physics (ICTP), is
open-source and extensively employed in regional climate studies, contributing to the establishment of a com-
prehensive Regional Climate Research Network (RegCNET) (Giorgi et al., 2006). The model can be applied to
all regions of the globe (Giorgi et al., 2012) and is moving into a fully-coupled regional Earth system model
framework through coupling with the ocean (Turuncoglu et al., 2013; Artale et al., 2010), lake (Small et al.,
1999), aerosol (Solmon et al., 2006), dust (Zakey et al., 2006), chemistry (Shalaby et al., 2012), hydrology
(Coppola et al., 2003), land surface processes (Oleson et al., 2008). Of specific interest for our study, Shalaby et
al. (2012) added a radiatively interactive gas-phase chemical module (CBM-Z) to RegCM4, generating
RegCM-Chem, in which atmosphere physics and chemistry are fully coupled.
**2.2.1 Aerosol Mechanisms**
The RegCM model integrates a simplified aerosol framework, enabling the simulation of sulfate, black
carbon (BC), organic carbon (OC), sea salt, and desert dust. The model specifies an external mix of aerosols and
accounts for the influence of horizontal advection, turbulent diffusion, vertical transport, emissions, dry and wet
deposition, and gas-liquid transition on aerosol concentration (Solmon et al., 2012; Giorgi et al., 2012; Zakey et
al., 2006). The secondary organic aerosol scheme VBS (volatile basis set) has also been introduced into the
model to further improve RegCM-Chem's simulation of tropospheric aerosols (Yin et al., 2015). The model in-
corporates the ISORROPIA thermodynamic equilibrium scheme to describe the formation process of secondary
inorganic salts, thus enhancing the model's capability to simulate secondary inorganic aerosols (Li et al., 2016b).
The further addition of bioaerosols was carried out by Liu (Liu et al., 2016).
**2.2.2 Gas phase chemical mechanism**
RegCM4-Chem includes the CBM-Z (Carbon Bond Mechanism-Z) atmospheric chemistry mechanism
(Zaveri and Peters, 1999). The CBM-IV mechanism, recognized for its widespread use, serves as the basis for
CBM-Z (Gery et al., 1989) and was developed to balance simulation accuracy and computational speed. Both
CBM-IV and CBM-Z categorize volatile organic compounds (VOCs) into groups dependent on their carbon



bond formation and use lumped species to represent each group. However, CBM-Z includes additional species
and reactions compared to CBM-IV, which are crucial for simulating typical urban environments and long-term
simulations at regional to global scales. Enhancements in CBM-Z include (1) specific representation of stable
alkanes; (2) updated parameters for higher alkanes; (3) separation of olefins into two categories based on differ-
ing reactions; (4) addition of peroxy alkane self-reactions significant in low-NOx, such as remote regions; (5)
incorporation of reactions among alkanes, peroxyacyl radicals, and $NO_3$, which are crucial nocturnally; (6) in-
clusion of long-lived organic nitrates and peroxides; and (7) refinement of isoprene and its peroxy radical chem-
istry. Collectively, these updates to the CBM-Z chemistry mechanism enhance the model's ability to more accu-
rately simulate long-lived VOCs and address the atmospheric chemistry transition from urban to rural settings.
**2.2.3 Radiation scheme**
RegCM4 adopts the CCM3 radiation scheme, which uses the delta-Eddington approximation for solar
spectral radiation and accounts for the attenuation effect of atmospheric components such as $O_3$, $H_2O$, $CO_2$, $O_2$
on solar radiation (Kiehl et al., 1996). The CCM3 radiation scheme, implemented in RegCM4, extends from 0.2
to 5 µm, and is segmented into 18 bands. It uses the cloud scattering and absorption parameter scheme, and
cloud optical characteristics. As cumulus clouds form, the cloud optical characteristics stretch from the cloud
base up to the cloud top, and the radiation calculations assume random overlap. It is assumed in the model that
the cloud thickness is equivalent to that of the model's vertical layers, with distinctive cloud water and ice
contents assigned to high, middle, and low clouds (Slingo, 1989).
**2.2.4 Photolysis rate**
Meteorological conditions and chemical input fields determine the photolysis rate, with most variables
dynamically produced by the RegCM's modules and updated every 3-30 minutes. $SO_2$ and NOx, inverted from
the US standard atmosphere's vertical profile, are model-defined. Owing to the computational demands of
precise photolysis rates from the Tropospheric Ultraviolet–Visible Model (TUV) method (Madronich and
Flocke, 1998) and eight data stream spherical harmonics discretization, a look-up table and interpolation method
are adopted. Considering the significant impact of clouds on the photolysis rate, it becomes crucial to adjust the
cloud amount. Here we use the cloud optical depth information for each grid cell within the model. As the
absorption and scattering of ultraviolet radiation by clouds reduce the photolysis rate inside and below the cloud



while enhancing it above the cloud, the correction value for the photolysis rate under clear sky conditions de-
pends on the position to the cloud layer. Hence, cloud height and optical depth are necessary for the photolysis
rate computation (Chang et al., 1987).
**2.2.5 Deposition Processes**

In the model, dry deposition serves as the principal removal process for trace gases, with the deposition

velocity being determined by three categories of resistance: aerodynamic, quasi-laminar sublayer, and surface
resistance, encompassing soil and vegetation absorption. The latter is inclusive of both stomatal and nonstomatal
absorption. The dry deposition module, taken from the CLM4 surface scheme, covers 29 gas-phase species and
comprises 11 types of land cover. To enhance the accuracy of the daily variation in dry deposition simulation,
both stomatal and nonstomatal resistances are accounted for in the dry deposition scheme. The calculation of all
deposition resistances is performed within the CLM land surface model (Wesely, 1989). Wet deposition uses the
MOZART global model's wet deposition parameterization scheme (Emmons et al., 2010; Horowitz et al., 2003),
including 26 gas-phase species in CBM-Z, and the wet deposition amount is based on the simulated precipita-
tion.
**2.3 Descriptions of the YIBs model**

The YIBs model, pioneered by Yale University, integrates plant physiological mechanisms to simulate how

photosynthesis, respiration, and other physiological processes respond to environmental drivers such as radia-
tion, temperature, and moisture. Moreover, YIBs simulates the carbon cycle both regionally and globally (Yue
and Unger, 2015). For example, its simulation of terrestrial carbon flux closely matches ground flux observa-
tions and satellite-derived data in diverse geographical areas such as the United States and China (Yue and
Unger, 2017; Yue et al., 2017).
**2.3.1 The main processes in YIBs**

In the YIBs model, eight distinct Plant Functional Types (PFTs) are incorporated, encompassing evergreen

coniferous forest, evergreen broad-leaved forest, deciduous broad-leaved forest, shrub forest, tundra, C3 grass-
land, C4 grasslands, and crops. The model employs the Michaelis–Menten enzyme-kinetics scheme for simulat-
ing plant photosynthesis (Farquhar et al., 1980), and the total photosynthesis ($A_{tot}$) of leaves is affected by Ru-





bisco enzyme activity ($J_c$), electron transfer rate ($J_e$), and photosynthetic product (triose phosphate) transport
capacity ($J_S$) limitation.

### 2.3.2 Canopy Radiation Scheme

A multilayer canopy radiation transmission scheme is adopted in YIBs for canopy radiation transmission
(Spitters et al., 1986), consisting of a radiation transfer model based on the total leaf area index, extinction coef-
ficient, and vegetation height. The entire vegetation canopy is usually divided into 2 to 16 layers, and the spe-
cific number of layers can be automatically adjusted according to the height of the canopy.

### 2.3.3 Biogenic Volatile Organic Compound Emission Scheme

Differently from the traditional MEGAN scheme, the YIBs model applies a biogenic volatile organic com-
pound (BVOC) emission scheme on a leaf scale, which is better suited to describe the photosynthesis process in
vegetation (Guenther et al., 1995). This introduces an effect of plant photosynthesis on BVOC emissions which
is more closely related to the real physiological process of vegetation. The intensity of leaf BVOC emission de-
pends on the rate of photosynthesis under electron transfer rate limitation, leaf surface temperature, and intra-
cellular $CO_2$ concentration.

### 2.3.4 Ozone Damage Protocol

When tropospheric ozone enters plants through stomata, it can directly damage plant cell tissues, thereby
slowing the photosynthesis rate and further weakening the carbon sequestration capacity of vegetation. The
YIBs model incorporates the semi-mechanistic parameterization scheme to delineate ozone's effect on plants
(Sitch et al., 2007).

### 2.4 Descriptions of the RegCM-Chem-YIBs model

### 2.4.1 Coupling between RegCM-Chem and YIBs

The integrated RegCM-Chem-YIBs model, an enhancement to the original RegCM-Chem, introduces $CO_2$
as an atmospheric constituent, incorporating its source-sink dynamics, transport, and diffusion processes. At-
mospheric $CO_2$ concentration is primarily influenced by atmosphere-ocean $CO_2$ exchange flux, biomass com-
bustion emissions, fossil fuel emissions, and terrestrial ecosystem $CO_2$ flux. The model prescribes fossil fuel





emissions, biomass combustion emissions, and atmosphere-ocean $CO_2$ fluxes, while the terrestrial ecosystem
$CO_2$ fluxes are computed in real time via the coupled YIBs terrestrial ecosystem model.

Within the coupled model system, meteorological variables (including temperature, humidity, precipitation,

radiation, etc.) and atmospheric pollutant concentrations ($O_3$ and $PM_{2.5}$) generated by RegCM-Chem are incor-
porated into the YIBs model every six-minute intervals. YIBs then simulates vegetation physiological processes
such as photosynthesis and respiration, computing land surface parameters including $CO_2$ flux, BVOC, and
stomatal conductance. These outputs from the YIBs are subsequently integrated back into the RegCM-Chem
model, modulating atmospheric composition ($CO_2$, $O_3$, and $PM_{2.5}$) and atmospheric variables (atmospheric tem-
perature, humidity, and circulation), thereby describing the interplay of climate, chemical, and ecological pro-
cesses.
**2.4.2 Model input data**

The input data of RegCM-Chem-YIBs mainly includes four categories: surface data, initial boundary data,

anthropogenic emission data and $CO_2$ surface flux data, which are detailed below.

(1) Surface data include surface vegetation cover type, terrain, and leaf area index. Land cover type infor-

mation is obtained from the MODIS and AVHRR satellites, employing the classification scheme suggested by
Lawrence and Chase (Lawrence and Chase, 2007), which uses MODIS data to preliminarily distinguish forest,
grassland, bare soil, etc., and combine this with AVHRR data to make a detailed forest classification. The dataset
contains a total of 16 different vegetation functional types. To align with the classification conventions of the
YIBs model, the original 16 vegetation functional types were converted into the corresponding 8 types recog-
nized by the YIBs model. The results are shown in Figure S1.

(2) Initial and boundary data include initial and boundary conditions of meteorological variables and at-

mospheric chemical composition. Here we use ERA-Interim reanalysis meteorological data, a product from the
European Center for Medium-Range Weather Forecasts (ECMWF) created through four-dimensional variational
assimilation. The data is on 37 vertical levels, with a horizontal resolution of 0.125°×0.125°, and time resolution
of 6 hours. Data for Sea Surface Temperature (SST) is provided by the weekly averaged Optimum Interpolation
SST product (OI_WK) of the National Oceanic and Atmospheric Administration (NOAA) (Reynolds et al.,
2002). The initial and boundary conditions of atmospheric chemical components (e.g. $O_3$), come from simula-
tions carried out with the global chemistry model MOZART (Emmons et al., 2010; Horowitz et al., 2003). In



addition, the initial and boundary conditions for $CO_2$ species come from the CarbonTracker global carbon as-
similation system (Peters et al., 2007) developed by NOAA Earth System Research Laboratory ESRL (Earth
System Research Laboratory), which uses the ensemble Kalman filter algorithm to assimilate ESRL greenhouse
gas observations and $CO_2$ observation data provided by the network of collaborating institutions worldwide. The
assimilated data includes not only conventional fixed-site observations but also mobile monitoring data such as
aircraft and ships. Since 2007, yearly updated carbon assimilation products are provided by CarbonTracker, de-
livering global $CO_2$ three-dimensional concentration data products every three hours. In this study, we utilized
the CT2019 product, updated in 2019, spanning a period from January 1, 2000 to March 29, 2019.

(3) Anthropogenic emission data include precursors of ozone and particulate matter such as NOx, VOC,

BC, OC, etc. The MIX Asian anthropogenic emission inventory developed by the Tsinghua University is used
(Li et al., 2017b), which integrates the results of the emission inventories of various regions in Asia. The emis-
sions in China come from China's multi-scale emission inventory MEIC (Multi-resolution Emission Inventory
for China) and the high-resolution $NH_3$ emission inventory developed by Peking University. The anthropogenic
emissions in India come from the Indian local emission inventory developed by ANL (Argonne National Labor-
atory), while the anthropogenic emissions in South Korea come from the CAPSS (The Korean local emission
inventory developed by the Policy Support System), and the man-made emissions in other regions are provided
by the REAS (Regional Emission inventory in Asia) emission inventory version 2.1. The anthropogenic emis-
sions of major pollutants in the simulated area are shown in Figure S2.

(4) Data pertaining to fossil fuel $CO_2$ emissions are sourced from the MIX Asian anthropogenic emission

inventory with a monthly time resolution. $CO_2$ emissions resulting from biomass burning are derived from the
FINN (Fire Inventory from NCAR) inventory (Wiedinmyer et al., 2011) developed by the National Center for
Atmospheric Research. The FINN inventory has a daily time resolution. The model's ocean-atmosphere $CO_2$
exchange flux is obtained from the carbon flux product of the CarbonTracker assimilation system, constructed
with the global atmospheric transport model TM5 and assimilating $CO_2$ observation data via an ensemble Kal-
man filter algorithm. This provides global $1° \times 1°$ resolution $CO_2$ exchange flux data between the ocean and the
atmosphere updated every three hours. The emissions are detailed in Figure S3.
**3 Model Application**
**3.1 Model setup**



To evaluate the performance of RegCM-Chem-YIBs we carried out a one-year simulation starting from
December 1st, 2015, through December 31st, 2016. The initial month is used as spin-up period, and thus it is not
included in the analysis. The simulation domain is centered at 36°N, 107°E, and covers a considerable part of
East Asia, including China, Japan, the Korean Peninsula, and Mongolia, along with significant parts of India and
Southeast Asia (Figure S4). The horizontal grid spacing is 30 km and we use 14 levels in the vertical, reaching
up to 50 hPa. Section 2.4.2 provides a comprehensive description of the model input data.
**3.2 Climate simulations in East Asian**
Given the importance of the climate for the East Asia region, we first present an assessment of the simula-
tion for the climate 2016 by comparison with the ERA-Interim data. The simulated temperature, specific
humidity, and wind fields at varying altitudes and seasons compared well with the reanalyzed data (Figure S5~
Figure S9), especially temperature and specific humidity, while a tendency to overestimate wind speed is
observed at the near surface and 850 hPa levels. The fields at 500 hPa show very close agreement with
reanalysis data, indicating a strong mid-atmosphere forcing by the boundary conditions, while the simulated
circulation patterns near the surface and at 850 hPa in summer tend to deviate more from the driving reanalysis.
The simulated circulation patterns in the other seasons are basically consistent with the reanalysis data.
Table 1 reports a number of statistical metrics of comparison between simulated and reanalysis
meteorological variables at different heights. Correlation coefficients (R) range from 0.95 to 0.98 for tempera-
ture, 0.71 to 0.97 for longitudinal wind, 0.81 to 0.92 for latitudinal wind, and 0.91-0.92 for specific humidity,
indicating a general good consistency between model and driving data, in line with previous studies (Zhuang et
al., 2018; Zhou et al., 2014; Wang et al., 2010).
**Table 1.** Statistical indicators for comparison between model simulation results and reanalysis data

| Heights | Statistical index | Air Temperature(K) | Longitudinal wind (m/s) | Latitudinal wind (m/s) | Specific humidity (kg kg$^{-1}$) |
|---|---|---|---|---|---|
| 500 hpa | R | 0.98 | 0.97 | 0.92 | 0.91 |
| | MB | 0.15 | 0.35 | -0.03 | 0.00015 |
| | RMSE | 0.93 | 0.75 | 0.51 | 0.00019 |
| 850 hpa | R | 0.96 | 0.77 | 0.85 | 0.94 |
| | MB | -0.98 | 0.38 | 0.15 | -0.00066 |





| | | | | | |
|---|---|---|---|---|---|
| | RMSE | 1.1 | 1.08 | 0.59 | 0.00077 |
| Near sur-face | R | 0.95 | 0.71 | 0.81 | 0.92 |
| | MB | -1.21 | 0.33 | 0.23 | -0.00098 |
| | RMSE | 1.35 | 0.59 | 0.54 | 0.00112 |

(Correlation coefficients (R), mean biases (MB), and root mean square error (RMSE))
The magnitude of surface radiation flux directly determines the rates of photosynthesis in vegetation. For
verification purposes, model surface solar fluxes were compared with data on solar energy at the surface
retrieved from the Clouds and the Earth's Radiant Energy System (CERES) satellite, which has a 1° × 1°
horizontal and monthly temporal resolution. Figure S10 shows the simulated surface net shortwave radiation in
different seasons and comparison with observational data. The model tends to overestimate surface net
shortwave radiation in spring and winter over India and summer over North China (Yin et al., 2014). Overall,
the simulated surface net shortwave radiation agrees well with the CERES satellite retrieval results, capturing
the spatial distribution and seasonal fluctuation patterns of surface shortwave radiation. The simulation findings
from our study are consistent with earlier research regarding surface net shortwave radiation (Han et al., 2016).
In conclusion, RegCM-Chem-YIBs demonstrates a good performance in simulating the climatological
features of the East Asia atmospheric circulations, effectively reproducing the spatial distribution and seasonal
variations of temperature, specific humidity, and radiation.
**3.3 Simulations of** $PM_{2.5}$**,** $O_3$ **and** $CO_2$
In this section, we compare simulated $PM_{2.5}$ and $O_3$ concentrations against observational data from 366
stations provided by the China National Environmental Monitoring Center. The geographical distribution of the
simulated annual mean near-surface daily $PM_{2.5}$ and maximum daily 8-hour average (MDA8) $O_3$ concentration,
along with the observed values, are shown in Figure 2. Supplementary Figure S11 then compares in a scat-
ter-plot format the observation and simulation results. Both figures demonstrate that the model reproduces the
spatial distribution patterns of $PM_{2.5}$ and $O_3$, with a significant agreement between modeled and measured
values across all stations. The statistical indicators of simulated and measured surface $PM_{2.5}$ and $O_3$ levels are
shown in Table S1, showing a correlation between simulation and observations of $O_3$ and $PM_{2.5}$ of 0.74 and 0.65,
respectively. The simulated $O_3$ concentrations are generally lower than observed in the Fenwei Plain of China, a
discrepancy possibly attributable to uncertainties in the emission inventory for this region. In summary, the



RegCM-Chem-YIBs model demonstrates a good ability to capture the spatial distribution of observed
near-surface ozone and particulate matter, especially in highly polluted areas.

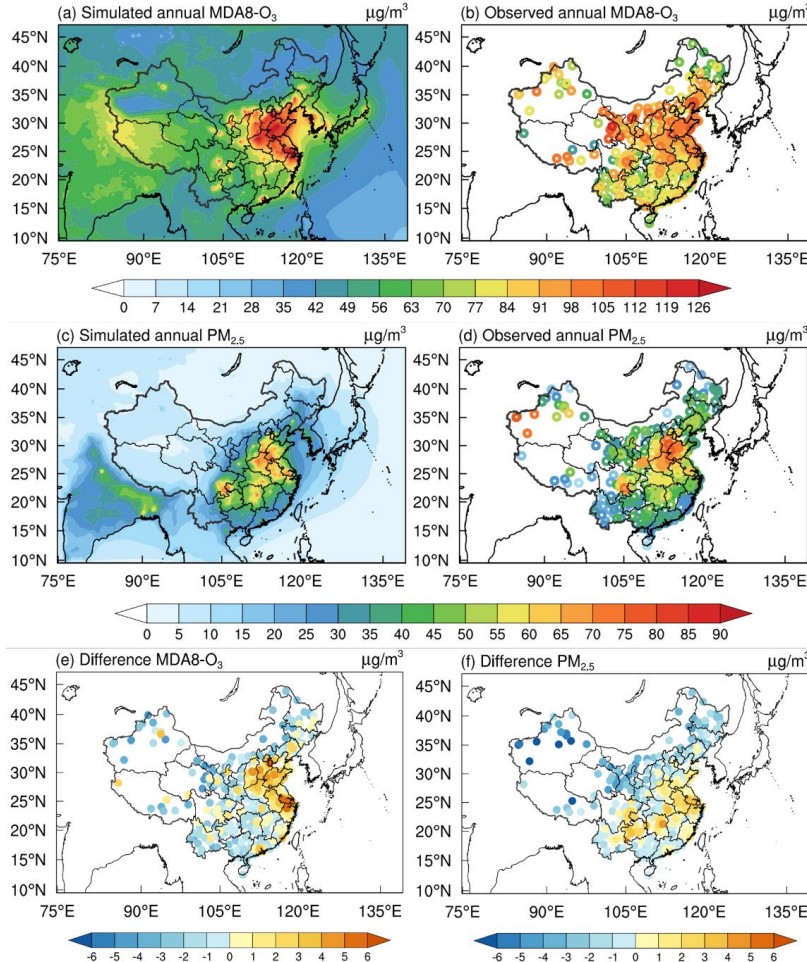


**Figure 2.** Simulation and observation comparison of (a, b) $O_3$ and (c, d) $PM_{2.5}$ and their differences (e, f) in China.
The differences are simulation minus observation. The colored circles in the figure represent station observations.
Units: $\mu g\ m^{-3}$.
Measured and calculated monthly mean $CO_2$ concentrations at six observation stations in East Asia from
the World data Center for Greenhouse Gases are shown in Figure 3. Information on the six sites is listed in Table
2. The simulated $CO_2$ concentration agrees well with observations, with correlation coefficients ranging from
0.89 to 0.97. However, in urban and coastal areas, the model performance deteriorates likely due to local emis-
sion fluctuations and errors in biogenic fluxes. Nevertheless, the model overall captures the seasonal variations



in $CO_2$ concentrations (Figure 3). This result likely stems from the complex relationship between biogenic and
fossil fuel emissions, which are known contributors to observed seasonal $CO_2$ patterns (Kou et al., 2015). A high
$CO_2$ mixing ratio (412.3 ppm) is observed at the TAP site, which is associated with strong local emissions. Fur-
ther analysis into the specific sources contributing to elevated $CO_2$ levels would provide valuable insights into
localized patterns of emissions and their effects on regional carbon cycle processes. The model's ability to re-
produce the geographical and seasonal $CO_2$ patterns serves as an illustration of its ability to capture the main
processes driving $CO_2$ dynamics. In summary, while discrepancies in urban or coastal areas highlight the chal-
lenges associated with capturing localized $CO_2$ dynamics, the model's overall performance and ability to repro-
duce geographical and seasonal $CO_2$ patterns demonstrates its usefulness in studying $CO_2$ dynamics at a regional
scale.
**Table 2.** Information on six $CO_2$ stations in East Asia and statistical indicators of observed and modeled $CO_2$.

| Sites | Latitude | Longitude | Elevation | Observations (ppm) | Simulations (ppm) | R | RMSE |
|-------|----------|-----------|-----------|--------------------|-------------------|-----|------|
| WLG | 36.29 | 100.90 | 3810 | 404.3 | 402.9 | 0.94 | 1.75 |
| TAP | 36.72 | 126.12 | 20 | 412.3 | 414.8 | 0.97 | 2.70 |
| UUM | 44.45 | 111.08 | 992 | 405.7 | 403.7 | 0.96 | 2.66 |
| LLN | 23.46 | 120.86 | 2867 | 406.0 | 407.2 | 0.93 | 1.63 |
| YON | 24.47 | 123.02 | 30 | 407.1 | 407.4 | 0.89 | 2.80 |
| HK | 22.31 | 114.17 | 65 | 407.9 | 409.7 | 0.92 | 15.67 |

(Correlation coefficients (R) and root mean square error (RMSE))

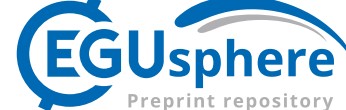

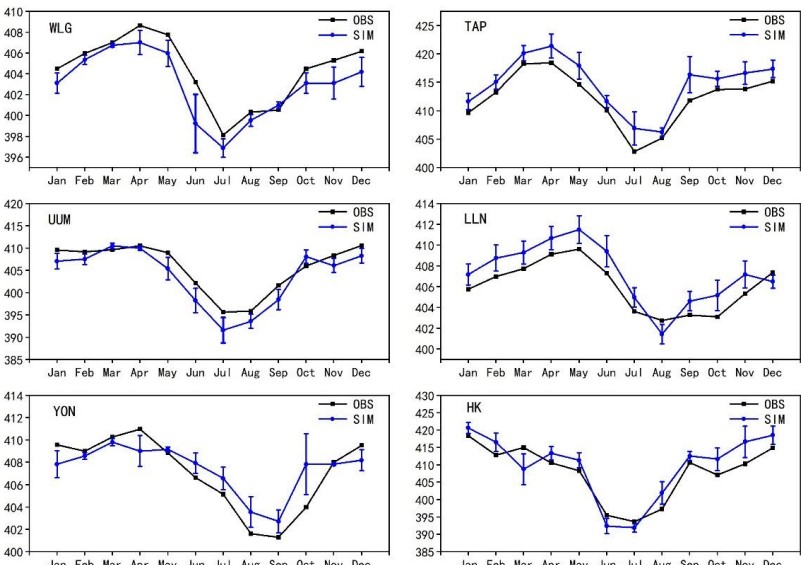


**Figure 3.** Modeled (blue) and observed (black) monthly mean $CO_2$ concentrations validated at six sites in East

Asia. Units: ppm.

The limitations of ground-based $CO_2$ observation stations, particularly their sparse spatial distribution, pose

challenges in obtaining high-resolution $CO_2$ data. To offset this limitation, data assimilation methods have been

implemented to ensure a coherent global distribution of atmospheric $CO_2$, effectively filling the void left by

sparse ground-based observations. Here we utilize the Carbon Tracker global carbon assimilation system

developed by the NOAA Earth System Research Laboratory (ESRL) to validate the simulated $CO_2$

concentrations (Peters et al., 2007). This comparison for the year 2016 is shown in Figure 4. The simulated $CO_2$

concentrations tend to be lower than observed in Northeastern India and Northeastern China, while they show a

better agreement with observations in other regions. These discrepancies can be traced back to factors such as

the underestimation of localized $CO_2$ emissions along with the effects of complex topography and circulation

patterns. However, the closer agreement in other regions suggests that the model effectively captures the

primary processes driving $CO_2$ concentrations.

Seasonal variations in the spatial distribution of $CO_2$ concentrations for 2016 are illustrated in supplemen-

tary Figure S12. The simulations show marked seasonal variations, with elevated concentrations in spring,

autumn, and lower values during summer. In northern regions, including Russia, Mongolia, and Northeast China,

the lowest near-surface $CO_2$ concentrations occur in summer. This pattern can be attributed to the enhanced

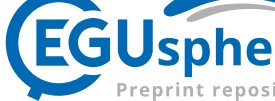

photosynthetic activity of terrestrial vegetation in summer, leading to enhanced atmospheric $CO_2$ sequestration.
Conversely, winter months are characterized by lower solar radiation fluxes and reduced vegetation
photosynthesis, resulting in relatively higher $CO_2$ concentrations. In specific regions, notably the eastern coastal
zones of China and South Korea, the seasonal pattern of $CO_2$ concentration is reduced, likely because of the
high levels of urbanization, dense population, and intense anthropogenic emissions in these areas. In contrast,
regions such as Yunnan, the southern side of the Qinghai-Tibet Plateau, and Southeast Asia exhibit consistently
low $CO_2$ concentrations during summer because of significant vegetation sinks in these densely vegetated areas.
An increase in $CO_2$ concentrations can be observed over these regions during spring due to local forest fires and
straw-burning processes, which release substantial amounts of $CO_2$ into the atmosphere (Chuang et al., 2014).

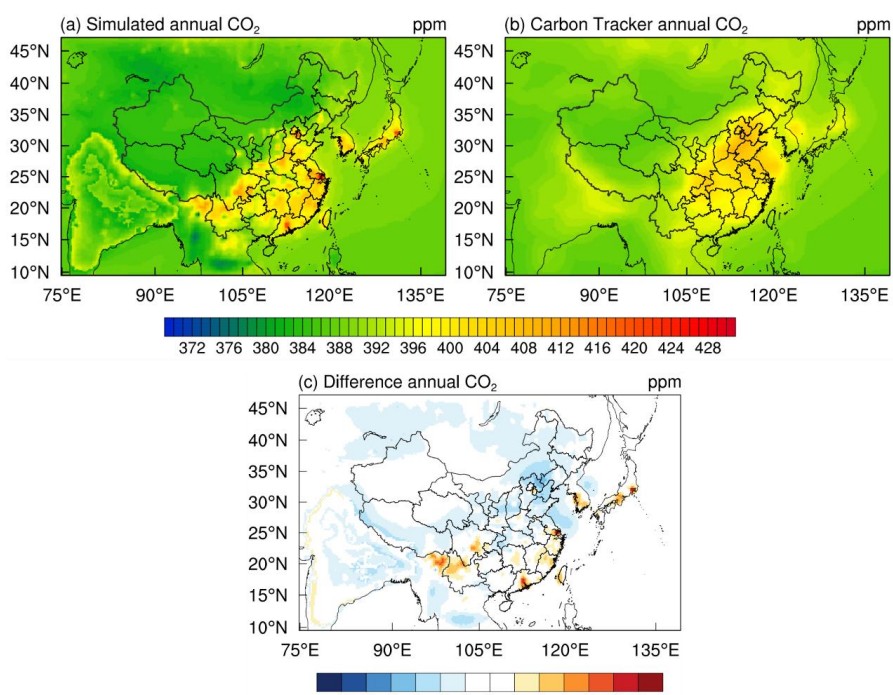


**Figure 4.** Evaluation of simulated $CO_2$ (a) using Carbon Tracker products (b) and their difference (c) in 2016. The
differences are simulation minus observation. Units: ppm.
**3.4 Simulations of carbon fluxes in terrestrial systems**
Our assessment of GPP and NPP uses the MOD17A3 Collection 6, a global product originating from
MODIS satellite observations. GPP data include 8-day values with a resolution of 500 meters, as produced in



MOD17A2H Version 6 based on radiation use efficiency theory. Such data can be used as input to computations
of terrestrial carbon and energy flows, water cycling processes, and vegetation biogeochemistry. Moreover, the
MOD17A3H Version 6 product provides information on annual NPP, also on a resolution of 500 meters. All
8-day Net Photosynthesis (PSN) products (MOD17A2H) from a particular year are combined to derive annual
NPP values (He et al., 2018; Madani et al., 2014; Running, 2012).

Figure 5 (a, b, e) shows the geographical distribution of the mean GPP in 2016 from the model simulations

and MODIS products. RegCM-Chem-YIBs effectively captures the observed spatial GPP features, with high
values mostly over Southwest, Central, and Southeastern China, areas characterized by deciduous broad-leaf
and evergreen coniferous forests (Figure S1). The annual average GPP simulated by RegCM-Chem-YIBs is
higher than observed over   Southwest and Central China by 6.8% and 12.7%, respectively. The annual average
simulated GPP over China is 6.18 Pg C $yr^{-1}$, which is about 7.56% higher than the GPP in MODIS.

Figure 6 (a) and Table S2 show the scatter plots of the simulated annual average GPP on each model grid

point compared with MODIS. A correlation coefficient of 0.91 and root mean square error of 0.4 kg C $m^{-2}$ $yr^{-1}$ is
found,   reflecting an overall good simulation by the model. Compared with the results obtained from the global
model NASA ModelE2–YIBs (Yue and Unger, 2017), the GPP value estimated here compares better with the
MODIS product, which may also be attributed to the higher spatial resolution of the regional system. Moreover,
our GPP results are also in line with earlier findings, such as from Li (Li et al., 2013b) who estimated an annual
average GPP over China of 6.04 Pg C $yr^{-1}$ based on the light energy utilization model EC-LUE.

Figure 5 (c, d, f) shows the spatial distribution of mean NPP for both the simulations and MODIS

products in 2016. NPP, similarly to GPP, exhibits a gradual reduction from southeast to northwest China. The
scatter plot comparing the simulated and MODIS annual average NPP across the model grid is illustrated in
Figure 6 (b). According to Table S2, a correlation coefficient of 0.87 is found between the simulated and
MODIS NPP, with a root mean square error of 0.22 kg C $m^{-2}$ $yr^{-1}$. Notably, the simulated NPP shows a distinct
underestimation over regions with higher NPP values. Compared with the MODIS NPP data products, the
annual average NPP simulated for the entire China region in 2016 is overestimated by approximately 8.64%,
mostly because of the model overestimate in Central China (16.6%).

Part of the reason for this result is the relatively simple treatment of the nitrogen deposition process in YIBs

(Yue and Unger, 2015). On the other hand, some studies have noted that due to the limitations of driving data
and algorithm parameters, the MODIS NPP products have some problems in China (Li et al., 2013b).



Furthermore, the NPP value estimated by the model over China is 3.21 Pg C yr$^{-1}$, in line with the mean value
(2.92 ± 0.12 Pg C yr$^{-1}$) found in previous 37 studies (Wang et al., 2017).

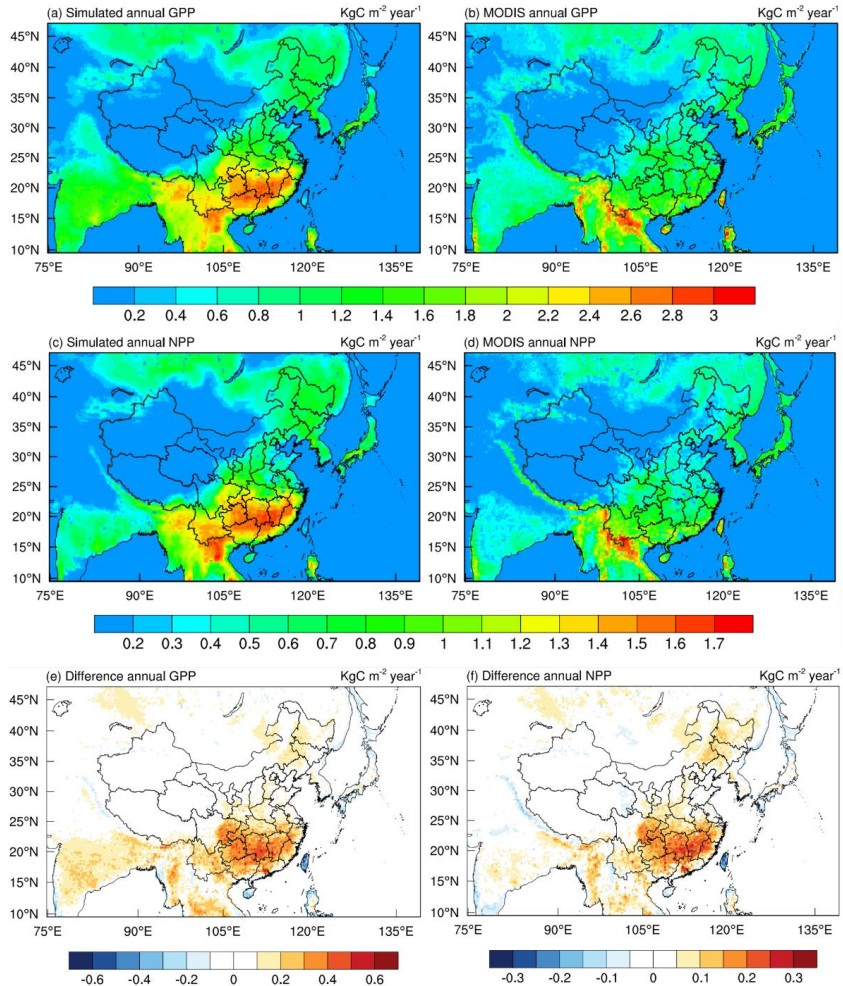


**Figure 5.** Spatial distribution of modeled (a, c) and MODIS (b, d), annual mean GPP (a, b) and NPP (c, d), and
their differences (e, f). The differences are simulation minus observation. Units: kg C m$^{-2}$ year$^{-1}$.



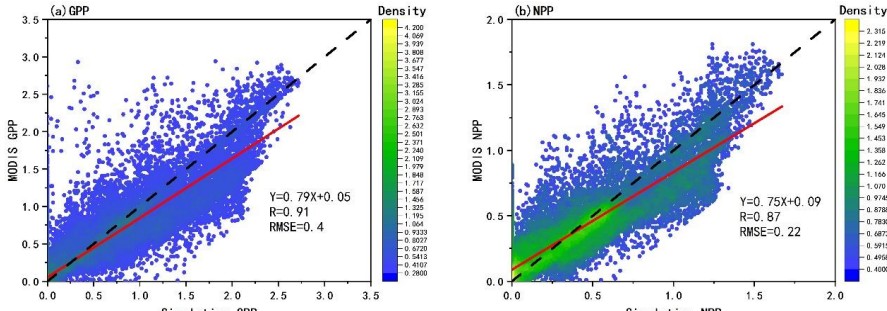


**Figure 6.** Density scatter plots of (a) GPP and (b) NPP for model simulations and inversion-based products for

2016. Units: kg C m$^{-2}$ year$^{-1}$.

Figure 7 and Figure 8 illustrate the seasonal fluctuations in GPP and NPP, as simulated for 2016 in East

Asia. Both GPP and NPP present pronounced seasonal variations, with negligible values during winter, and a

strong peak in summer. The winter minimum is attributable to limiting environmental factors such as reduced

solar radiation, lower temperatures, and suppressed photosynthetic activity by vegetation. Conversely, summer

shows the highest GPP and NPP values due to extended daylight hours, increased solar radiation, and

temperatures facilitating increased photosynthetic activity and vegetation metabolism.

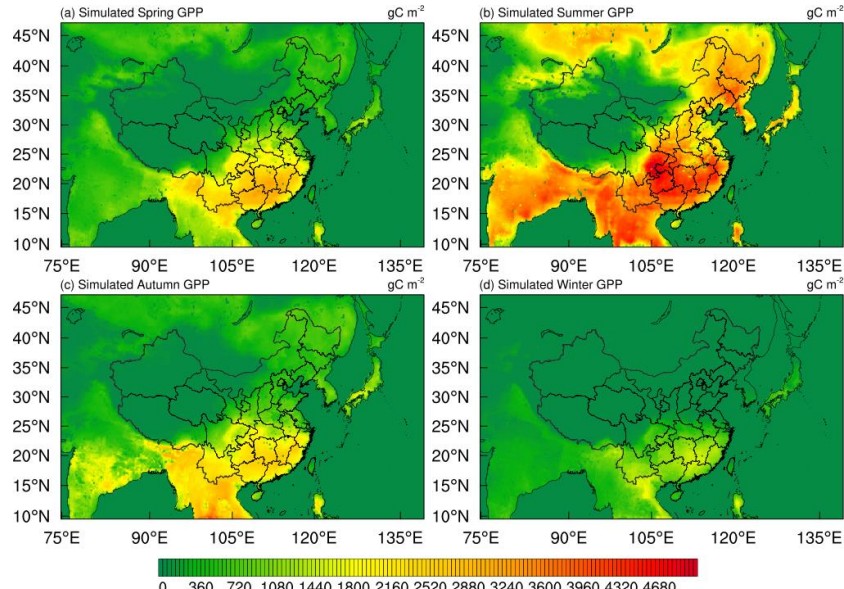

**Figure 7.** Spatial distribution of GPP simulated by model of spring(a), summer(b), autumn(c) and winter(d) in

2016. Units: g C m$^{-2}$





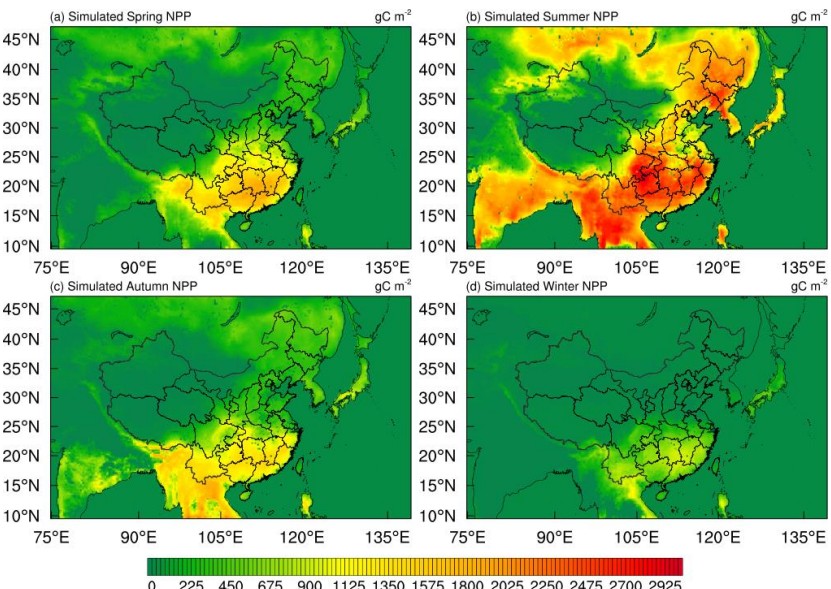

**Figure 8.** Spatial distribution of NPP simulated by model of spring(a), summer(b), autumn(c) and winter(d) in 2016. Units: g C m$^{-2}$

## 3.5 Simulations of other carbon-bearing species

The analysis of additional carbonaceous compounds such as BC, OC and carbon monoxide (CO), is crucial due to their considerable influence on climate and the carbon cycle. The spatial distribution of simulated BC for each season of 2016 is shown in Figure S13. BC concentrations are mainly centered in North China, Central China, the Sichuan Basin, Chongqing, and Northeast India, regions with a higher concentration of industrial and residential emission sources. BC displays a marked seasonal variation, with elevated levels in winter, possibly attributed to residential heating, more stagnant conditions, and reduced removal by precipitation.

Figure S14 then shows the spatial corresponding distribution of seasonal OC, which is also higher over North China, Central China, Sichuan and Chongqing, and Northeast India. Finally, Figure S15 reports the annual mean near-surface CO concentrations for observations and simulation data across the monitoring sites in China. While simulated CO concentrations agree well spatially with observations, the simulations produce higher values than observed in Central China, likely linked to uncertainties in emission inventories. Figure S16 presents the seasonal spatial distributions of CO, with simulated high values mostly localized in Sichuan-Chongqing and Central China, and a peak in winter.



**4 Conclusions**
Regional climate-chemical coupled models can be used to study the characteristics of regional-scale cli-
mate and pollutants, and is an important means to investigate the behavior of atmospheric pollutants and their
radiative climate effects. However, current coupled regional climate models describe the physiological process
of terrestrial vegetation relatively simply and do not consider the interaction between atmospheric pollutants
(such as $PM_{2.5}$ and $O_3$) and $CO_2$, as well as their impacts on terrestrial ecosystems.
To overcome this problem, in this work we coupled the YIBs biogeochemical model to the RegCM-CHEM
regional climate-chemistry model, and tested this coupled modeling system over a domain covering East Asia at
a 30 km horizontal grid spacing for the year 2016. The model output was validated against reanalysis data, ob-
servational data, and satellite remote sensing data, both for the atmosphere and the carbon cycle.
Our simulations show that the coupled RegCM-Chem-YIBs system can effectively reproduce the spa-
tio-temporal distribution of meteorological variables, atmospheric composition ($PM_{2.5}$, $O_3$, and $CO_2$) and terres-
trial carbon fluxes (GPP and NPP). Comparisons of the simulated temperature, longitudinal wind, latitudinal
wind, and specific humidity for different seasons with the driving ERA-Interim reanalysis data showed correla-
tion coefficients of 0.95-0.98, 0.71-0.97, 0.81-0.92, and 0.91-0.92, respectively. The correlation coefficients
between observed and simulated $O_3$ and $PM_{2.5}$ levels in China were 0.74 and 0.65, respectively, while the corre-
sponding correlations for $CO_2$ were in the range of 0.89 to 0.97. Comparison of the ecological parameters GPP
and NPP simulated in East Asia with the observed data showed correlation coefficients of 0.91 and 0.87, respec-
tively. In addition, in all cases, the seasonal variation of the different variables was captured by the model.
Therefore, we conclude that, overall, the RegCM-Chem-YIBs model demonstrates a good performance in simu-
lating the spatio-temporal distribution characteristics of regional meteorological characteristics, atmospheric
composition, and ecological parameters over East Asia.
In the future, we will continue to improve RegCM-Chem-YIBs in the following aspects. First, we will in-
vestigate the impact of $CO_2$ and $O_3$ inhomogeneity on radiation calculations by integrating temporally and spa-
tially varying concentrations derived from YIBs and Chem into the RegCM radiation module. This will enable
additional accurate computation of longwave radiation flux, improving the representation of the regional radia-
tion balance. Second, we intend to assimilate a module representing various chemical transformations happening
on the surfaces of aerosol particles. Finally, we will include the wet removal process of $O_3$. These advancements
will contribute to the refinement of RegCM-Chem-YIBs, enhancing our ability to investigate the interactions



462 between regional atmosphere, carbon cycle, and vegetation processes.

**Code and data availability**

464 The RegCM-Chem source code can be obtained from https://github.com/ICTP/RegCM (last access: 10 July

465 2023). The YIBs model code is available at https://github.com/YIBS01/YIBs_site (last access: 10 July 2023).

466 The input data and source code for RegCM-Chem-YIBs have been archived on Zenodo at

467 https://doi.org/10.5281/zenodo.8186164 (Xie and Wang, 2023). The CarbonTracker data are provided at

468 (https://gml.noaa.gov/ccgg/carbontracker/). The CERES surface radiation data are available at

469 (https://ceres.larc.nasa.gov/). WDCGG data are available at (https://gaw.kishou.go.jp/). CNEMC data are pro-

470 vided at (http://www.cnemc.cn/). MODIS data are available at (https://ladsweb.modaps.eosdis.nasa.gov/).

**Author contributions**

472 TW led the development of RegCM-Chem-YIBs with significant contributions from NX and XX. NX per-

473 formed the evaluation. NX, TW drafted the manuscript and all authors contributed to review and editing of the

474 manuscript.

**Competing interests**

476 The corresponding author has stated that all the authors have no conflicts of interest.

**Disclaimer**

478 Publisher's note: Copernicus Publications remains neutral about jurisdictional claims in published maps and

479 institutional affiliations.

**Acknowledgments**

481 This work was supported by the National Natural Science Foundation of China (42077192), the National Key

482 Basic Research & Development Program of China (2020YFA0607802), the Creative talent exchange program

483 for foreign experts in the Belt and Road countries, and the Emory University-Nanjing University Collaborative

484 Research Grant.



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
