# Peer review of "The Regional Climate-Chemistry-Ecology"

_EGUsphere, 2023_

## Author Comment (AC1)

**Response to reviewers on: The Regional Climate-Chemistry-Ecology Coupling Model RegCM-Chem (v4.6)-YIBs (v1.0): Development and Application**

**Nanhong Xie et al.**

We are grateful to the referees for their time and energy in providing helpful comments and guidance that have improved the manuscript. In this document, we describe how we have addressed the reviewer's comments. Review comments in black, responses in blue and text added/modified in manuscript in red.

=======================================================================

**RC1: 'Comment on egusphere-2023-1733', Anonymous Referee #1, 03 Jan 2024**

**General comments**:

In my opinion, it is valuable to investigate coupled interactions between the terrestrial carbon cycle, atmospheric chemistry, and climate change in regional scale. The manuscript entitled: "The Regional Climate-Chemistry-Ecology Coupling Model RegCM-Chem (v4.6)-YIBs (v1.0): Development and Application" presents valuable results which merit publication in egusphere after minor corrections.

**Response**: Thank you for your positive evaluations. We have carefully considered your suggestions and revised the paper accordingly.

**Specific comments:**

There are comments I suggest the authors should take into account: In 3.2 Climate simulations

in East Asian, the representativeness of statistical indicators for comparison between model simulation results and reanalysis data is not clear. The author did not provide the temporal resolution of statistical indicators such as Correlation coefficients (R), mean biases (MB), and root mean square error (RMSE).

**Response**: Thank you for your valuable feedback. We calculated statistical indicators such as correlation coefficient (R), mean bias (MB), and root mean square error (RMSE) using daily mean data. Although the meteorological variables temperature, wind speed and specific humidity in our coupled model simulation results are hourly, the reanalysis data ERA-Interim used for validation, are at 6-hour resolution (there are four times of data every day, which are 00,06,12 and 18 o'clock). To ensure consistency in the validation process between simulated results and reanalysis data, we first aggregated the data to daily mean values and then computed the corresponding statistical indicators (R, MB, RMSE). In the revised manuscript, additional clarification regarding the temporal resolution of these statistical indicators has been included in Section 3.2 (lines 334-337).

**Revised version**: "We first calculated the daily average of the meteorological variables, such as temperature, wind speed, and specific humidity, from the model simulation and reanalysis data, respectively. Then we calculate the corresponding statistical indicator correlation coefficient (R), mean deviation (MB), and root mean square error (RMSE) based on the daily averages. Table 1 reports a number of statistical metrics of comparison between simulated and reanalysis meteorological variables at different heights. Correlation coefficients (R) range from 0.95 to 0.98 for temperature, 0.71 to 0.97 for longitudinal wind, 0.81 to 0.92 for latitudinal wind, and 0.91-0.92 for specific humidity …"

---

## Author Comment (AC2)

**Response to reviewers on: The Regional Climate-Chemistry-Ecology Coupling Model RegCM-Chem (v4.6)-YIBs (v1.0): Development and Application**

**Nanhong Xie et al.**

We are grateful to the referees for their time and energy in providing helpful comments and guidance that have improved the manuscript. In this document, we describe how we have addressed the reviewer's comments. Review comments in black, responses in blue and text added/modified in manuscript in red.

===========================================================================

**RC2: 'Comment on egusphere-2023-1733', Anonymous Referee #2, 03 Jan 2024**

"This work built a new framework that couples terrestrial biosphere model and regional climate model. It is a novel and important investigation to explore the interaction between plants and climate change. Although the new model was evaluated using multi source data which showed a nice comparison, I think you should evaluate the ability of couple model through comparing the model results between couple and independent model (either biogenic model or climate model). We need to see if the coupled model works better than independent model. If so, how better? What is the dominant mechanism to improve the model perform through coupling? In addition, the description of couple method is not enough detail to help reader to understand this work. For example, meteorological variables and atmospheric pollutant concentrations generated by RegCM-Chem are incorporated into the YIBs model every six-minute intervals. Why do you determine the interval of six minutes? Another example, outputs from the YIBs are subsequently integrated back into the RegCM-Chem model, modulating atmospheric

composition and atmospheric variables. How often do those outputs integrate back? How can these outputs from YIBs model change the atmospheric variables? Directly or indirectly? How do you calculate the biogenic VOC emission flux, which is needed to state briefly although it was described in reference."

**We have divided the above comments into the following four specific comments based on content to facilitate better response and modification.**

**Specific comments:**

**Original comment 1#:** This work built a new framework that couples terrestrial biosphere model and regional climate model. It is a novel and important investigation to explore the interaction between plants and climate change.

**Response**: Thank you for your positive review of our research work.

**Original comment 2#**: Although the new model was evaluated using multi source data which showed a nice comparison, I think you should evaluate the ability of couple model through comparing the model results between couple and independent model (either biogenic model or climate model). We need to see if the coupled model works better than independent model. If so, how better? What is the dominant mechanism to improve the model perform through coupling?

**Response:** The reasons why we use multi source data verification instead of coupled mode and independent mode comparison verification are as follows:

(1) The coupling model has more advantage than the independent model. In the traditional model, meteorological, chemical and terrestrial ecological models are separately or in offline way. For example, after the meteorological field is completely calculated, they are used to drive the chemical model or terrestrial ecological model. In this case, the mutual feedback effects between meteorology, chemistry and vegetation cannot be considered. This study aims to develop a new tool that can be used to study the mutual feedback effects of climate, chemistry and terrestrial carbon cycle at regional scale, thus can realize online dynamic feedback, consider the process more comprehensively, and capture different systems more

comprehensively.

(2) It is difficult to compare the coupled model with the independent model because the coupled model has new output variables such as $CO_2$, which were not available in the previous independent model RegCM-Chem. In the development of coupling model, we first added $CO_2$ to RegCM-Chem and considered four kinds of source-sink processes, and then coupled it with the ecological model YIBs (see section 2.4.1), thus being able to consider the vegetation in YIBs dynamic sink process. In addition, our coupled model considers the damage to vegetation caused by the atmospheric chemical component $O_3$ (see section 2.3.4). The other independent model, YIBs, is based on static meteorological and $CO_2$ concentration inputs. Therefore, we believe that the coupled model and the independent model are not suitable for comparison due to large differences in input and output. In addition, we have verified the independent model through observational data in previous studies which can be referred to (Ma et al, 2023; Zhuang et al 2018; Yin et al, 2015; Li et al, 2016; Yue and Unger, 2015).

(3) The purpose of our verification based on multi-source data (reanalysis data, satellite data, site data, etc.) is to verify the capabilities of the coupled model from multiple different perspectives, such as climate, chemical and ecological variables. We believe this is the direct way to characterize the performance of a model. Our verifications show that the coupled model shows good performance in simulation of climate, chemistry, and ecology variables.

For the above reasons, we believe that the performance of the coupled model can be verified using the multi-source data.

**Original comment 3#:** In addition, the description of couple method is not enough detail to help reader to understand this work. For example, meteorological variables and atmospheric pollutant concentrations generated by RegCM-Chem are incorporated into the YIBs model every six-minute intervals. Why do you determine the interval of six minutes? Another example, outputs from the YIBs are subsequently integrated back into the RegCM-Chem model, modulating atmospheric composition and atmospheric variables. How often do those outputs integrate back? How can these outputs from YIBs model change the atmospheric variables? Directly or indirectly?

**Response:** Thank you for pointing this out. The decision to integrate meteorological variables and atmospheric pollutant concentrations into the YIBs model at six-minute intervals is based on a comprehensive consideration of several factors. Firstly, this six-minute interval is chosen to ensure alignment between the integration time step of the chemical module and the integration step of the YIBs model. This is crucial to prevent information distortion and ensure accurate representation of interactions between the two modules. Secondly, this time scale is selected considering the need to capture rapid variations in meteorological and chemical variables within short time frames. This synchronization and high spatiotemporal resolution are essential for accurately simulating dynamic atmospheric processes. The output of YIBs returns and adjusts for atmospheric and chemical variables as well for 6 minutes. Considering the complexity of chemical reactions and ecological processes, dynamic adjustments at short intervals enable the model to better capture transient interactions between ecology and the atmosphere. The choice of this adjustment frequency balances the representation of actual processes with computational efficiency, ensuring that simulation results are both accurate and efficient. YIBs outputs have both direct and indirect effects on atmospheric variables. Directly, $CO_2$ concentration undergoes significant changes due to the model's adjustments in dynamic vegetation sink processes. Indirectly, atmospheric components like $PM_{2.5}$, $O_3$, and others experience variations due to ecological feedback from vegetation. Additionally, the dynamic adjustments by YIBs influence atmospheric parameters such as temperature, humidity, and circulation, indicating the complex interactions between different components in the model. We have made detailed modifications to Chapter 2.4.1 in the revised manuscript (Lines 253-272) to help readers better understand this work.

**Revised version**: Within the coupled model system, meteorological variables (including temperature, humidity, precipitation, radiation, etc.) and atmospheric pollutant concentrations ($O_3$ and $PM_{2.5}$) generated by RegCM-Chem are incorporated into the YIBs model every six-minute intervals. This integration step is to be consistent with the integration time step of the chemistry module, thus maintaining synchronization between modules. Considering the complexity of chemical reactions and ecological processes, dynamic adjustments at short intervals enable the model to better capture transient interactions between ecology and the atmosphere. The choice of this adjustment frequency balances the representation of actual

processes with computational efficiency, ensuring that simulation results are both accurate and efficient. YIBs then simulates vegetation physiological processes such as photosynthesis and respiration, computing land surface parameters including $CO_2$ flux, BVOC, and stomatal conductance. These outputs from the YIBs are subsequently integrated back into the RegCM-Chem model every six-minute intervals, the intricacies of this integration process lead to significant changes in various environmental parameters. The major direct changes, prominently influencing the model's behavior, arise from alterations in $CO_2$ concentration. These changes are directly attributed to intricate physiological processes within the vegetation, including photosynthesis and respiration. The fluxes of $CO_2$ through these biological processes play a pivotal role in shaping the atmospheric composition. On the indirect front, the integration of YIBs outputs induces intricate variations in $PM_{2.5}$ and $O_3$ concentrations. These indirect changes are primarily orchestrated by shifts in BVOC emissions. The dynamic nature of these emissions contributes to the complexity of atmospheric chemistry, influencing the levels of $PM_{2.5}$ and $O_3$. Simultaneously, the integration process plays a crucial role in shaping the temporal variations of atmospheric temperature, humidity, and circulation. These changes over time are intricately linked to variations in land surface parameters. The interplay of these variables illustrates the dynamic feedback loops between climate, chemical composition, and ecological processes within the integrated model system.

**Original comment 4#:** How do you calculate the biogenic VOC emission flux, which is needed to state briefly although it was described in reference.

**Response:** We agree with you. Although there are several corresponding references, the descriptions of BVOC estimation are too simple. We will supplement the detailed calculation of BVOC emissions in Chapter 2.3.3 of the revised manuscript (Lines 202-222). In addition, we found that there may be such problems in the description of ozone damage to vegetation, so we also modified the relevant content of Chapter 2.3.4 about ozone damage to vegetation in the revised manuscript (Lines 227-242) to help readers understand the specific calculation process more clearly.

**Revised version**:

**2.3.3 Biogenic Volatile Organic Compound Emission Scheme**

[revised manuscript text omitted]